# Editing by Reconstruction: Background Preservation for Instruction-based Autoregressive Image Editing

## Abstract

Autoregressive (AR) editors have recently emerged as strong competitors to diffusion models for text-based image editing, yet they often introduce unintended changes in non-edited regions due to stochastic token sampling. We present ERec (Editing by Reconstruction), a background-preservation method that synchronizes sampling between reconstruction and editing and requires no additional fine-tuning. Concretely, we run a reconstruction path alongside the standard editing path and inject identical standard-Gumbel noise into both logits at every decoding step. This Gumbel-max procedure is multinomial-equivalent, so it keeps diversity while coupling the two chains: when the logits are similar (typically in background regions), token choices align; when they differ (true edit regions), choices diverge and editability is retained. After generation, a lightweight post-refinement localizes edits by combining distributional discrepancy with background confidence, followed by connectivity filtering and residual compositing to correct encoder quantization residuals. ERec requires no fine-tuning of the baseline, integrates seamlessly with top-$k$ or nucleus sampling, and adds negligible inference overhead. Experimental results show that it substantially improves background preservation while maintaining edit fidelity.

## 1 Introduction

Text-based image editing has long been a particularly important topic in generative AI (Mirza & Osindero, 2014; Li et al., 2025a; Chen et al., 2024). An effective image editing method should faithfully enact the text-specified change while minimally disturbing unrelated regions, preserving background and overall layout. Early approaches primarily relied on GAN inversion with CLIP-based guidance (Patashnik et al., 2021; Xia et al., 2021; Gal et al., 2022; Abdal et al., 2022). Recent advances in diffusion models (Ho et al., 2020; Song et al., 2021a) have substantially advanced generative modeling and, in turn, propelled text-based image editing to the forefront (Huang et al., 2025). Building on these models, most editing methods either leverage large-scale pretraining (Saharia et al., 2022; Esser et al., 2024) or adopt task-specific fine-tuning (Mokady et al., 2023; Zhang et al., 2023; Han et al., 2023; Shi et al., 2024). Most recently, autoregressive (AR) models, traditionally dominant in natural language processing (Achiam et al., 2023; Liu et al., 2024), have only more recently gained traction in visual synthesis (Ramesh et al., 2021; Esser et al., 2021).

In contrast to diffusion models, AR-based visual architectures align naturally with large language models (LLMs) by operating in the same next-token prediction framework over discrete token sequences, enabling tighter text–image integration and fine-grained token-level control (Chang et al., 2022). In particular, recent AR-based visual models (Tian et al., 2024; Xiao et al., 2025; Chen et al., 2025), such as LlamaGen (Sun et al., 2024) and VAR (Tian et al., 2024), have advanced image tokenization and transformer architectures, achieving performance competitive with diffusion models and demonstrating significant promise in visual generation. Building on this line of work, Mu et al. (2025) proposed an instruction-based AR image editing model EditAR, which fine-tunes LlamaGen as the backbone and establishes an instruction-based editing paradigm comparable to other instruction-driven diffusion models (Brooks et al., 2023; Zhang et al., 2024).

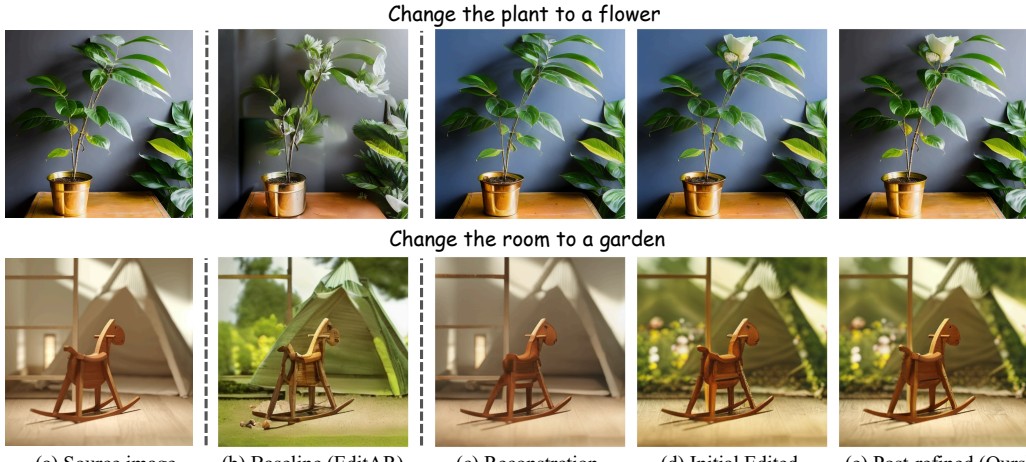

Change the plant to a flower

Change the room to a garden

(a) Source image  (b) Baseline (EditAR)  (c) Reconstrction  (d) Initial Edited  (e) Post-refined (Ours)

Figure 1: Illustration of uncontrolled background changes in AR editing and the workflow of our proposed ERec. (b) We adopt EditAR (Mu et al., 2025) as the baseline, which often introduces unexpected alterations in the background during editing (e.g., deformation of the flowerpot and distortion of the wooden horse). (c) Our ERec leverages reconstruction as guidance. (d) It synchronizes sampling between reconstruction and editing, keeping background tokens on the reconstruction trajectory. (e) Finally, by comparing with the reconstruction tokens (with probabilities), ERec identifies the true editing regions and achieves robust background control.

Despite these advances, visual AR models are still criticized for their limited ability to preserve non-edited regions consistency, a weakness that becomes particularly evident in instruction-based editing methods (Hu et al., 2025; Wu et al., 2025). This limitation stems from their next-token generation paradigm, where each prediction during inference is sampled from a conditional distribution given both the input tokens (from input image and instruction text) and the previously generated tokens. While this sampling mechanism is the source of AR models' diversity and editing flexibility, it can also lead to incorrect predictions in regions that should remain unchanged, causing cascading alterations in the non-edited regions, as shown in Fig. 1(b). This challenge remains underexplored, especially in the context of instruction-based AR editing.

In this paper, building on the EditAR as baseline, we propose a background-preservation method, Editing by Reconstruction (ERec), which requires no further fine-tuning of the baseline. Conceptually, regions susceptible to change under editing exhibit behavior similar to reconstruction: their conditional distributions are closely aligned. Nevertheless, independent sampling can steer these regions along divergent token trajectories, inducing extensive background drift. To mitigate this, we inject shared standard-Gumbel perturbations into the logits of both reconstruction and editing at every decoding step, thereby implementing multinomial sampling via the Gumbel-max trick (Jang et al., 2016). This preserves output diversity while synchronizing trajectories in background regions wherever the distributions coincide, without constraining edited areas (Fig.1(c,d)). After the initial generation, we compute, for each token, the Jensen–Shannon divergence between the reconstruction and editing distributions and combine it with token-level confidence to delineate the true edit mask; we then restore non-edited regions with the original tokens with pixels, yielding robust background preservation (Fig.1(e)).

The main contribution of this paper can be summarized as follows:

- We introduce ERec, which uses a reconstruction pass as background guidance and shares the same standard-Gumbel perturbations between reconstruction and editing at every step. This keeps sampling multinomial-equivalent and synchronizes token choices in non-edited regions while leaving true edits unconstrained, yielding strong background preservation.

- We design a refinement procedure that jointly leverages distributional discrepancy between reconstruction and editing and token confidence to localize the true edit regions, then restores background areas with the original input tokens with pixel residuals.

- ERec requires no further fine-tuning of the baseline editing AR model and no multi-round inference, yielding substantial background-preservation gains while maintaining edit fidelity.

## 2 RELATED WORKS

### 2.1 TEXT-GUIDED IMAGE EDITING

Text-guided image editing aims to modify image content according to a semantic prompt while preserving irrelevant background regions. Early GAN-based approaches (Mirza & Osindero, 2014; Xia et al., 2021; Gal et al., 2022) optimize in the GAN latent space, often with CLIP-based guidance (Patashnik et al., 2021; Abdal et al., 2022), but are constrained by the capacity and domain coverage of the pretrained generator, yielding limited realism and poor faithfulness out of distribution. Diffusion-based editing has since become mainstream. A distinguishing property of diffusion models is the (approximate) one-to-one link between an image and its latent/noise trajectory $z$ under the forward noising process; consequently, most pipelines first invert a real image to its latent state and then perform conditional generation with a new prompt, enabling principled inversion and reconstruction-consistent edits. Inversion can be deterministic (e.g., DDIM inversion or flow-matching trajectories) (Song et al., 2021a; Lipman et al., 2022) or stochastic via SDE/DDPM-style paths (Song et al., 2021b; Ho et al., 2020); editing is then guided by cross-attention control (Hertz et al., 2022) or test-time optimization (Tumanyan et al., 2023; Lu et al., 2023; Hertz et al., 2024). Training-based variants bypass test-time inversion by learning on paired data (Brooks et al., 2023), and practical refinements improve reconstruction alignment and efficiency (Mokady et al., 2023; Miyake et al., 2025). In contrast, autoregressive models generate images token by token without an explicit invertible latent tied to the input, making faithful inversion and strict background preservation more challenging.

### 2.2 AUTOREGRESSIVE IMAGE GENERATION AND EDITING

**Autoregressive vision models.** Unlike diffusion models' iterative denoising, AR models synthesize images by predicting the next token in a discrete sequence. Early autoregressive approaches modeled generation at the pixel or token level (van den Oord et al., 2016; 2017; Esser et al., 2021) or through masked prediction (Chang et al., 2022). Recent large-scale AR models (Sun et al., 2024; Tian et al., 2024; Wang et al., 2024; Luo et al., 2024; Xiao et al., 2025; Chen et al., 2025) have further improved image tokenization and adopted LLM-style Transformer backbones. These advances enable AR models to produce high-quality images competitive with diffusion models, while supporting flexible conditioning from semantic text (Ramesh et al., 2021; Yu et al., 2022; Ding et al., 2022) or structural cues such as edges and depth (Li et al., 2024; 2025b). These capabilities provide a strong foundation for AR-based image editing.

**AR-based image editing.** AR-based image editing remains comparatively underexplored, especially with respect to background preservation. Mu et al. (2025) propose EditAR, which fine-tunes a strong LlamaGen prior to follow natural-language edit instructions, serving as a competitive AR editing baseline. Hu et al. (2025) introduce a training-free anchoring strategy that implicitly locks scene structure and curbs background drift during localized edits. Most recently, Dao et al. (2025) propose Discrete Noise Inversion, inspired by diffusion-based inversion, which searches for a Gumbel noise realization that exactly reconstructs the input token sequence and treats the recovered noise as a discrete latent for editing. Although we also employ Gumbel sampling, our use and objective are different: instead of searching for a specific noise code of the latent $z$ to locate the source image in the distribution, we simply synchronize Gumbel draws between reconstruction and editing. Because logits in non-edit regions remain close to their reconstruction counterparts, reapplying the same draws faithfully simulates multinomial sampling for background preservation while leaving edit regions unconstrained. In summary, despite the natural alignment of AR editors with LLM-style token interfaces and their fine-grained token-level control, the literature consistently reports vulnerability to error accumulation and background drift—issues our method is designed to mitigate.

## 3 METHOD

The proposed ERec targets background-preserving editing for the instruction-based AR editor baseline EditAR (Mu et al., 2025), which has two stages: (1) background token trajectory alignment via a dual-path inference (reconstruction and editing) with synchronized Gumbel-max sampling; and (2) post-refinement that leverages distributional discrepancy and source-token confidence with simple spatial connectivity and residual compositing. We first review the AR editing paradigm in Section 3.1, then detail inference in Section 3.2 and post-refinement in Section 3.3.

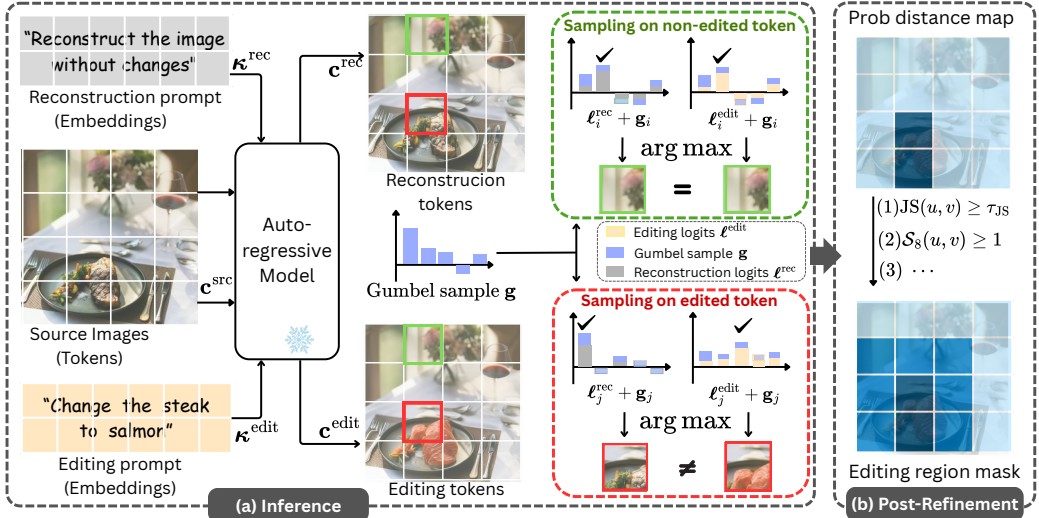

Figure 2: Pipeline of ERec. (a) Inference. Background token choices are traced via a reconstruction path, and identical standard-Gumbel perturbations are injected into the logits of both reconstruction and editing at each step. When the logits are similar, token selections in non-edit regions (green boxes) are aligned; where the logits differ substantially, tokens in true edit regions (red boxes) remain unconstrained. (b) Post-refinement. The edit mask is derived by combining token-wise probability discrepancy with spatial connectivity, after which non-edited regions are restored to the original input (tokens and pixels). Best viewed in color.

## 3.1 PRELIMINARIES

The instruction-based AR editor EditAR (Mu et al., 2025), instantiated with LlamaGen (Sun et al., 2024), is fine-tuned on interventional triplets and casts image editing as next-token prediction conditioned on the source image and the instruction.

**Tokenization.** Let $\mathbf{X} \in \mathbb{R}^{H \times W \times 3}$ be the input RGB image. A pre-trained VQ-VAE (van den Oord et al., 2017; Razavi et al., 2019) encodes $\mathbf{X}$ into a latent map $\mathbf{z} \in \mathbb{R}^{H_\ell \times W_\ell \times n_z}$. Each vector $\mathbf{z}_{u,v} \in \mathbb{R}^{n_z}$ is quantized to its nearest codebook entry from $\mathcal{C} = \{\mathbf{e}_1, \ldots, \mathbf{e}_V\}$, yielding a discrete index $c_{u,v} \in \{1, \ldots, V\}$. Using the flattened index $i \in \{1, \ldots, N\}$ with $N = H_\ell W_\ell$, the token is $c_i$. The source token sequence is $\mathbf{c}^{\mathrm{src}} \in \mathcal{V}^N$ with $\mathcal{V} = \{1, \ldots, V\}$. An instruction $\mathcal{P}$ is processed by a pretrained tokenizer and text encoder (e.g., T5 (Raffel et al., 2020b) used in LlamaGen) to obtain a text representation $\boldsymbol{\kappa}^{\mathrm{text}1}$.

**Conditioning & decoding.** For instruction-based editing, conditioning is provided by concatenating the text representation and the source token sequence, $[\boldsymbol{\kappa}^{\mathrm{text}}, \mathbf{c}^{\mathrm{src}}]$, and the AR model outputs a target token sequence $\mathbf{c}^{\mathrm{tgt}} = (c_1^{\mathrm{tgt}}, \ldots, c_N^{\mathrm{tgt}})$ of the same length as $\mathbf{c}^{\mathrm{src}}$. After inference, $\mathbf{c}^{\mathrm{tgt}}$ is mapped back to a latent code map via codebook lookup, $\hat{\mathbf{z}}_{u,v} = \mathbf{e}_{c_{u,v}^{\mathrm{tgt}}}$ with $\{\mathbf{e}_k\}_{k=1}^V$ denoting the VQ-VAE codebook vectors, and subsequently decoded by the VQ-VAE decoder to produce the edited image thereby realizing pixel-space image editing $\mathbf{X}^{\mathrm{tgt}}$.

**AR factorization.** Decoding proceeds autoregressively: tokens are generated sequentially and each prediction conditions on the already produced prefix. Accordingly, by the chain rule, the conditional likelihood factorizes as

$$p\big(\mathbf{c}^{\mathrm{tgt}} \mid \boldsymbol{\kappa}^{\mathrm{text}}, \mathbf{c}^{\mathrm{src}}\big) = \prod_{i=1}^{N} p\big(c_i^{\mathrm{tgt}} \mid \mathbf{c}_{<i}^{\mathrm{tgt}}, \boldsymbol{\kappa}^{\mathrm{text}}, \mathbf{c}^{\mathrm{src}}\big), \qquad \mathbf{c}_{<i}^{\mathrm{tgt}} = (c_1^{\mathrm{tgt}}, \ldots, c_{i-1}^{\mathrm{tgt}}). \qquad (1)$$

**Sampling strategy.** Token-level sampling directly governs background preservation and instruction fidelity. Common choices include greedy ($\arg\max$), full multinomial, and truncated variants such as top-$k$ and nucleus (top-$p$). While $\arg\max$ tends to preserve non-edited content, it often suppresses the stochasticity needed to realize edits, leading to degeneration or instruction failure (Welleck et al.,

---

[1]We omit vocabulary-to-embedding conversion details on the text side as they are orthogonal to our analysis.

2019; Holtzman et al., 2020). Conversely, sampling from a large candidate set (large $k$ or full multinomial) increases off-target tokens in background regions; due to exposure-bias accumulation, local mistakes can cascade into broader background drift (Bengio et al., 2015; Arora et al., 2022).

## 3.2 RECONSTRUCTION-GUIDED INFERENCE

We now describe stage (1) in Fig. 2(a). In AR-based image editing, background drift arises from the interplay of model interpretation and sampling randomness. Empirically, regions susceptible to edit-induced drift also exhibit high variability under the reconstruction prompt. Exploiting this, we synchronize the sampling noise between reconstruction and editing: at each decoding step we inject the same standard-Gumbel perturbation vector to both logits, making the two samplers stochastically identical given their logits. This procedure couples background choices where the logits are similar, while leaving true edit regions unconstrained where the logits differ.

**Tokenization with Reconstruction.** ERec augments the standard editing generation with an auxiliary reconstruction pipeline. That is, we run two pipelines in parallel at inference: (i) editing, conditioned on the original editing instruction, and (ii) reconstruction. Given a source image and an editing instruction, we additionally define a reconstruction prompt phrased as *"reconstruct the image without any changes"* to elicit reconstruction. Denote the text embeddings of the editing and reconstruction prompts by $\kappa^{\text{edit}}$ and $\kappa^{\text{rec}}$, respectively. Let $\mathbf{c}^{\text{src}}$ be the tokenized source image, and let $\mathbf{c}^{\text{rec}}$ and $\mathbf{c}^{\text{edit}}$ be the token sequences produced by the pre-trained AR model under the reconstruction and editing prompts with elements $c_i^{\text{rec}}$ and $c_i^{\text{edit}}$, respectively.

**Record-and-replay with shared Gumbel noise.** In AR-based image generation, output diversity and editability stem from per-token multinomial sampling. Yet this randomness is hard to align across runs: even with the same seed and nearly identical logits, the editing and reconstruction pipelines condition on different prompts, yielding different token choices on background. To expose and synchronize this randomness, we adopt *Gumbel-max* sampling, i.e., sampling by adding i.i.d. Gumbel noise to logits. This lets us record the Gumbel draws during reconstruction and replay the same draws at background tokens during editing. At position $i$, ERec evaluates two logit vectors

$$\boldsymbol{\ell}_i^{\text{rec}} = f_\theta(\mathbf{c}_{<i}^{\text{rec}}; \kappa^{\text{rec}}, \mathbf{c}^{\text{src}}), \qquad \boldsymbol{\ell}_i^{\text{edit}} = f_\theta(\mathbf{c}_{<i}^{\text{edit}}; \kappa^{\text{edit}}, \mathbf{c}^{\text{src}}), \tag{2}$$

where $f_\theta$ is the pre-trained autoregressive decoder. Applying a softmax over the vocabulary converts these logits into categorical distributions, denoted by $\mathbf{p}_i^{\text{rec}}$ and $\mathbf{p}_i^{\text{edit}}$.

We draw a single Gumbel noise vector $\mathbf{g}_i \sim \text{Gumbel}(0, 1)^{|\mathcal{V}|}$, with the draw deterministically keyed by $(i, \text{seed})$. This ensures that $\mathbf{g}_i = (g_{i,1}, \ldots, g_{i,|\mathcal{V}|})$ is identical across the reconstruction and editing pipelines and remains independent of their conditioning. We then add the same noise to the logits and select by argmax for both pipelines:

$$c_i^{\text{rec}} = \arg\max_{k \in \mathcal{V}}\{\ell_{i,k}^{\text{rec}} + g_{i,k}\}, \qquad c_i^{\text{edit}} = \arg\max_{k \in \mathcal{V}}\{\ell_{i,k}^{\text{edit}} + g_{i,k}\}. \tag{3}$$

This Gumbel-max procedure is exactly equivalent in distribution to sampling from the multinomial (categorical) defined by the softmax of the logits; it merely makes the randomness explicit, hence recordable and replayable (see Appendix B for the equivalence). Sharing the same $\mathbf{g}_i$ induces a stable coupling: when the two logits are similar (typically in non-edited regions), the argmaxes coincide with high probability and the chains align; when they differ (edited regions), the choices naturally diverge, preserving editability while reproducing the reconstruction-time behavior elsewhere.

**Integration with top-$k$ sampling.** In contrast to existing methods (Hu et al., 2025; Dao et al., 2025), ERec does not explicitly steer the model toward copying the original input tokens during inference stage. Instead, reconstruction serves as guidance to record background token choices and to synchronize sampling there by reusing the same Gumbel draws. The mechanism is plug-and-play with common sampling strategies such as top-$k$ and nucleus (top-$p$) sampling: when using top-$k$ or top-$p$, we first form the candidate set $\mathcal{K}_i$ from the original logits, then add Gumbel noise to the logits restricted to $\mathcal{K}_i$ and take an argmax. By the truncation corollary (in Appendix B), this is exactly equivalent to sampling from the renormalized distribution over $\mathcal{K}_i$; thus ERec does not alter the underlying sampling probabilities. As a result, the edit region remains fully editable while the background replays the reconstruction trajectory without additional intervention. The per-token record-and-replay step with shared Gumbel noise is given in Algorithm 2.

## 3.3 POST-REFINEMENT

During inference, we steer the AR model to keep background regions aligned with the reconstruction trajectory. Building on this, we apply a lightweight post-refinement to identify the actual edited regions. We rely on two token-wise signals: (i) a distributional distance between the reconstruction and editing predictions (e.g., JS divergence), and (ii) the probability assigned to the source token as a proxy for background confidence (negative log-likelihood, NLL). We index locations either by a flattened index $i$ or by 2D coordinates $(u, v)$ on the $H_\ell \times W_\ell$ latent grid; let $M \in \{0, 1\}^{H_\ell \times W_\ell}$ denote the edit mask, where $M_{u,v} = 1$ marks an edit location and $M_{u,v} = 0$ marks background.

**JS-based seeding.** Under shared Gumbel noise, reconstruction and editing tokens in background regions are likely to follow the same sampling chain with similar logits. We thus measure the discrepancy between $\mathbf{p}_{u,v}^{\text{rec}}$ and $\mathbf{p}_{u,v}^{\text{edit}}$ using the Jensen–Shannon (JS) divergence (Lin, 1991):

$$\text{JS}(u, v) := \tfrac{1}{2} \text{KL}_2\big(\mathbf{p}_{u,v}^{\text{rec}} \,\|\, \mathbf{m}_{u,v}\big) + \tfrac{1}{2} \text{KL}_2\big(\mathbf{p}_{u,v}^{\text{edit}} \,\|\, \mathbf{m}_{u,v}\big), \quad \mathbf{m}_{u,v} = \tfrac{1}{2}\big(\mathbf{p}_{u,v}^{\text{rec}} + \mathbf{p}_{u,v}^{\text{edit}}\big), \quad (4)$$

where $\text{KL}_2$ uses base-2 logarithms, so $\text{JS} \in [0, 1]$. JS is symmetric and bounded and smaller values indicate higher similarity. Given a JS threshold $\tau_{\text{JS}}$, we first threshold the JS map to obtain a coarse edit mask and then apply a one-hop 8-connected dilation:

$$M_{u,v}^{(0)} := \mathbf{1}\left\{ \max_{(a,b) \in \mathcal{N}_8[u,v]} \text{JS}(a, b) \geq \tau_{\text{JS}} \right\}, \quad \mathcal{N}_8[u, v] := \{(a, b) : |a - u| \leq 1, |b - v| \leq 1\}, \quad (5)$$

which preserves edit coverage and enforces local spatial consistency.

**NLL-based refinement.** The JS-seeded mask covers most edits but also may include background false positives. Empirically, reconstruction tokens are more volatile than editing ones in background regions (editing concentrates attention near the instruction, while reconstruction is more diffuse), so we refine using per-location NLL on the source token:

$$\text{NLL}_{u,v}^{\text{edit}} = -\ln\big(\mathbf{p}_{u,v}^{\text{edit}}[c_{u,v}^{\text{src}}]\big), \qquad \text{NLL}_{u,v}^{\text{rec}} = -\ln\big(\mathbf{p}_{u,v}^{\text{rec}}[c_{u,v}^{\text{src}}]\big). \qquad (6)$$

Let $\tau_{\text{low}}$ and $\tau_{\text{high}}$ be thresholds hyper-parameters for NLL. Mark a location as background if either condition holds:

$$\text{(a) } \text{NLL}_{u,v}^{\text{edit}} \leq \tau_{\text{low}} \quad \text{or} \quad \text{(b) } \text{NLL}_{u,v}^{\text{rec}} \geq \tau_{\text{high}}. \qquad (7)$$

Intuitively, (a) accepts locations that the editing pass confidently assigns to the background, preventing spurious changes caused by occasional reconstruction errors while (b) covers cases with jointly high uncertainty, potentially arising from token-selection cascades; we conservatively treat such locations as background. Denote the indicator by $S_{u,v} \in \{0, 1\}$, with $S_{u,v} = 1$ indicating a background location. We then shrink the initial editing mask by abstaining on background positions:

$$M_{u,v} = M_{u,v}^{(0)} \wedge \big(1 - S_{u,v}\big). \qquad (8)$$

**Pixel-level refinement.** After obtaining the binary edit mask $M \in \{0, 1\}^{H_\ell \times W_\ell}$ (1 = edit), we lock the background in token space:

$$c_{u,v}^{\text{edit}} \leftarrow \begin{cases} c_{u,v}^{\text{src}}, & M_{u,v} = 0, \\ c_{u,v}^{\text{edit}}, & M_{u,v} = 1, \end{cases} \qquad \hat{\mathbf{X}}_{\text{edit}} = \text{Dec}\big(\mathbf{c}^{\text{edit}}\big), \qquad (9)$$

where Dec is the pre-trained VQ-VAE decoder and $\mathbf{c}^{\text{src}}$ are source image tokens. With the mask in hand, we further mitigate VQ-VAE decoding losses by aligning at the pixel level, akin to background-preserving compositing widely used in image editing and recent text-driven editing pipelines (Hur & Roth, 2019; Avrahami et al., 2022):

$$\hat{\mathbf{X}}_{\text{src}} = \text{Dec}\big(\mathbf{c}^{\text{src}}\big), \quad \mathbf{R} = \mathbf{X} - \hat{\mathbf{X}}_{\text{src}}, \quad \mathbf{Y} = \hat{\mathbf{X}}_{\text{edit}} + \big(\mathbf{1} - M_\uparrow\big) \odot \mathbf{R}, \qquad (10)$$

where $M_\uparrow$ is the binary edit mask upsampled to image resolution, $\mathbf{1}$ is the all-ones matrix of matching dimension, and $\odot$ denotes element-wise multiplication. Here $\mathbf{Y}$ denotes the final refined output: the edited image $\hat{\mathbf{X}}_{\text{edit}}$ corrected with residual details $\mathbf{R}$ in background regions. This residual compositing recovers high-frequency textures and subtle color cues lost to quantization of VQ-VAE, while leaving the edited regions unchanged. Our post-refinement is detailed in Algorithm 3 and the full ERec is in Algorithm 1.

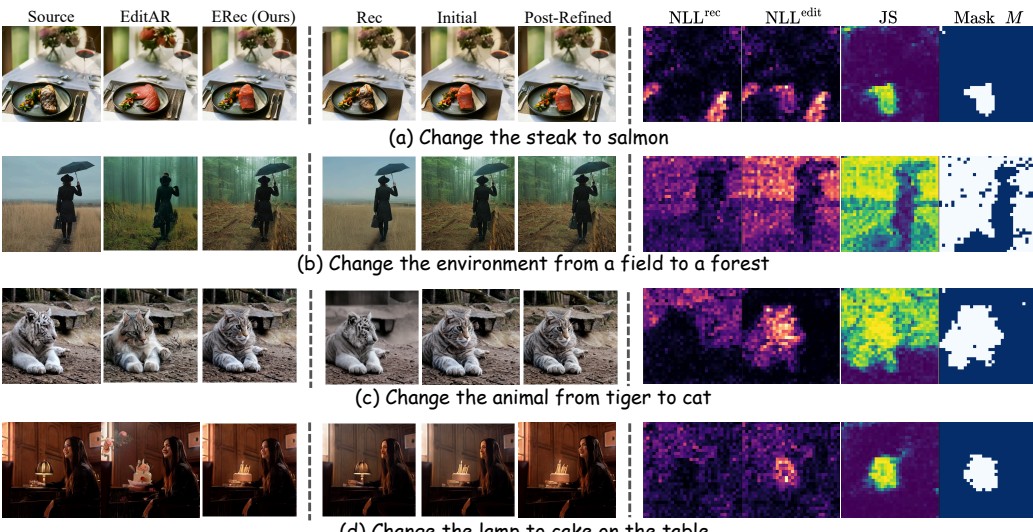

Figure 3: Qualitative results and intermediate process visualization. Left: Visual comparisons between the baseline EditAR and our method ERec. Middle: ERec reconstruction, initial editing, and post-refined output (without pixel alignment). Right: Inference-time diagnostics: NLL of reconstruction, NLL of editing, JS distance between their token distributions, and the estimated background mask. Brighter colors indicate higher NLL/JS values.

Table 1: Comparison of the proposed ERec, baseline EditAR and various diffusion-based approaches.

| Method | T2I Model | Structure Distance ↓ | Background Preservation | | | | CLIP Similarity | |
|---|---|---|---|---|---|---|---|---|
| | | | PSNR ↑ | LPIPS ↓ | MSE ↓ | SSIM ↑ | Whole ↑ | Edited ↑ |
| Prompt-to-Prompt | SD1.4 | 88.46 | 16.80 | 270.38 | 241.89 | 69.93 | 26.70 | 21.43 |
| Null-text Inversion | SD1.4 | 18.42 | 25.68 | 77.70 | 42.92 | 85.71 | 24.55 | 20.73 |
| Pix2pix-zero | SD1.4 | 59.43 | 19.71 | 193.44 | 147.19 | 76.48 | 23.56 | 19.76 |
| MasaCtrl | SD1.4 | 34.20 | 21.59 | 124.35 | 83.60 | 81.31 | 22.90 | 18.52 |
| PnPInversion | SD1.5 | 24.81 | 22.16 | 114.15 | 74.07 | 81.81 | 25.56 | 21.50 |
| InstructPix2Pix | SD1.5 | 67.49 | 19.69 | 164.27 | 235.62 | 76.98 | 23.37 | 20.48 |
| MGIE | SD1.5 | 53.46 | 20.62 | 131.13 | 205.09 | 79.55 | 22.67 | 19.58 |
| EditAR | LlamaGen | 38.46 | 21.43 | 117.80 | 132.51 | 75.09 | 24.03 | **21.45** |
| ERec (Ours) | LlamaGen | **26.52** | **26.63** | **76.50** | **75.24** | **88.20** | **24.13** | 21.44 |

# 4 EXPERIMENTS

**Implementation details.** Our backbone is EditAR (Mu et al., 2025), which builds on LlamaGen (Sun et al., 2024) and is fine-tuned on triplet-paired editing datasets (source image, instruction, target), including SEED-Data-Edit-Unsplash (Ge et al., 2024) and PIPE dataset (Wasserman et al., 2025), to enable instruction-based image editing. We follow the default hyperparameters of EditAR (see Appendix C.1 for the full configuration) with top-$k = 1000$ unless stated otherwise. For the hyperparameters in ERec, we set the JS threshold to $\tau_{\text{JS}} = 0.7$, and the NLL thresholds to $\tau_{\text{low}} = 3.0$ and $\tau_{\text{high}} = 10$. The reconstruction prompt is fixed as: *"reconstruct the image without changes"*.

**Evaluation protocol and metrics.** We evaluate on standard instruction-based image editing benchmark, PIE-Bench (Ju et al., 2024), which contains 700 examples covering 10 editing types. Our method takes the source image and editing instruction from the dataset as input and predicts the edited target with the reconstruction guidance. Following prior work, we assess both reconstruction fidelity and text-alignment of the edited content with the dataset's provided foreground masks. For quantitative evaluation, we follow common practice. Background preservation (outside the edit mask) is measured by Structure Distance, PSNR, LPIPS, MSE, and SSIM. Prompt–image consistency is assessed with the CLIP similarity score between the target prompt and the generated image. Additional details about datasets and metric computation are provided in Appendix C.2.

Table 2: Comparison under different top-$k$ samplers. Bold marks the better score within each pair.

| TopK | Methods | Structure Distance ↓ | Background Preservation | | | | CLIP Similarity | |
|---|---|---|---|---|---|---|---|---|
| | | | PSNR ↑ | LPIPS ↓ | MSE ↓ | SSIM ↑ | Whole ↑ | Edited ↑ |
| 1 | Baseline | 28.53 | 23.40 | 81.42 | 162.34 | 78.58 | 23.06 | 20.42 |
| | ERec (Ours) | **13.83** | **35.63** | **34.46** | **46.18** | **99.60** | **23.17** | **20.47** |
| 512 | Baseline | 37.25 | 21.57 | 113.62 | 132.65 | 75.51 | 24.12 | 21.43 |
| | ERec | **23.09** | **30.08** | **62.93** | **55.77** | **95.11** | **24.13** | **21.45** |
| 16K | Baseline | 41.08 | 21.15 | 127.86 | 138.20 | 73.96 | 24.03 | 21.61 |
| | ERec | **30.80** | **27.96** | **85.86** | **82.54** | **86.60** | **24.24** | **21.63** |

## 4.1 EXPERIMENTAL RESULTS

**Quantitative comparison.** Table 1 compares our method ERec with the AR baseline EditAR (Mu et al., 2025) and a suite of diffusion-based approaches, including Prompt-to-Prompt (Hertz et al., 2022), Null-text Inversion (Mokady et al., 2023), Pix2Pix-Zero (Parmar et al., 2023), MasaCtrl (Cao et al., 2023), PnP Inversion (Ju et al., 2024), InstructPix2Pix (Brooks et al., 2023), and MGIE (Fu et al., 2023). Our primary focus is the comparison with EditAR: quantitatively, ERec improves background consistency and overall similarity to the target while maintaining strong editing performance within the edited regions, and it is competitive with diffusion-based approaches.

**Qualitative comparison.** Figure 3 presents qualitative comparisons on several representative cases. EditAR often introduces stochastic changes in non-edited regions, whereas ERec better preserves background content. On closer inspection, although ERec can also make occasional sampling errors during reconstruction, these errors are propagated to the editing run with a small JS distance and are therefore identified as non-edited areas (background in (a) and foreground in (b)). In case (c), even when reconstruction deviates, our $NLL_{low}$–based criterion still isolates the true edit region. In case (d), because ERec estimates an edit/background mask, we perform pixel-space alignment to reduce VQ-VAE quantization artifacts—most visible on fine details such as facial expressions and text. More examples are provided in Figure 6. Overall, ERec robustly localizes edit regions and preserves background.

**Top-$k$ sampling.** Our method integrates naturally with existing sampling strategies. Using top-$k$, the results in Table 2 show that: whether using argmax, full multinomial or any intermediate $k$, our approach consistently achieves better background preservation while maintaining fidelity in the edited regions compared to the baseline. For additional $k$ values, please refer to Table 5 in the Appendix.

Table 3: Runtime.

| Methods | Time(s) |
|---|---|
| Baseline | $33.86 \pm 0.09$ |
| ERec | $37.19 \pm 0.06$ |

**Inference time.** ERec can parallelize the reconstruction pipeline in batch to avoid redundant passes. Table 3 gives the per-image runtime on a single NVIDIA H100, averaged over 10 trials.

## 4.2 ABLATION STUDY

In this section, we assess the contribution of each component in ERec, as summarized in Table 4.

**Direct Replacement.** We begin with a straightforward variant, *Direct Replacement*: under shared-Gumbel sampling, whenever the JS divergence between reconstruction and editing distributions falls below a threshold, we directly substitute the edited token with the *source* token, i.e., $c^{edit} \leftarrow c^{src}$. In our experiments, we apply this rule with $JS < 0.3$. From the results, although this strategy yields strong background preservation, it severely degrades editing ability: the model tends to output source tokens without applying the intended edits, leading to poor target fidelity.

**Ablation on main modules.** In the middle block of Table 4, we further dissect the effect of ERec's main components. (1) Independent vs. shared Gumbel. We replace shared-Gumbel sampling with independent Gumbel draws while keeping post-refinement and pixel alignment unchanged. This substitution markedly degrades background preservation compared with shared Gumbel. Figure 4 gives a visualization: with independent Gumbel, both the edited and reconstruction trajectories make occasional background mistakes, but because their samples differ, the JS distance becomes high and these locations are (wrongly) flagged as edit regions—expanding the predicted edit mask. In contrast, with shared Gumbel the same background errors occur in both trajectories, yielding

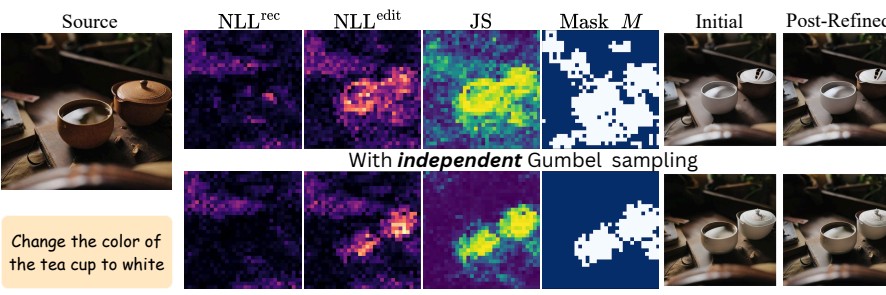

Figure 4: Comparison of *independent* vs. *shared* Gumbel sampling. Sharing Gumbel draws between reconstruction and editing suppresses sampling-induced JS spikes, yielding cleaner edit masks and stronger background preservation.

Table 4: Ablation study (Top-$k$=2048).

| Method | Structure Distance ↓ | Background Preservation | | | | CLIP Similarity | |
|---|---|---|---|---|---|---|---|
| | | PSNR ↑ | LPIPS ↓ | MSE ↓ | SSIM ↑ | Whole ↑ | Edited ↑ |
| EditAR (Baseline) | 39.56 | 21.42 | 120.25 | 138.24 | 74.83 | 24.10 | 21.45 |
| Direct Replacement | 17.11 | 25.08 | 66.70 | 57.59 | 79.59 | 23.57 | 20.73 |
| ERec (Ours) | 26.21 | 25.99 | 80.19 | 75.11 | 87.56 | 24.09 | 21.44 |
| ERec w/o Shared Gumbel | 28.12 | 24.44 | 84.04 | 95.43 | 83.78 | 24.06 | 21.45 |
| ERec w/o Post-Refinement | 42.38 | 20.75 | 128.82 | 198.63 | 74.04 | 23.98 | 21.51 |
| ERec w/o Pixel Alignment | 30.71 | 22.68 | 100.65 | 98.37 | 76.98 | 24.06 | 21.42 |
| *Base*: + JS | 27.32 | 26.18 | 76.39 | 130.52 | 87.82 | 23.97 | 21.26 |
| + dilation | 35.41 | 23.51 | 101.03 | 165.58 | 82.65 | 24.02 | 21.43 |
| + $NLL_{low}$ | 30.97 | 25.02 | 90.01 | 134.16 | 84.33 | 24.07 | 24.42 |
| + $NLL_{high}$ (ERec) | 26.21 | 25.99 | 80.19 | 75.11 | 87.56 | 24.09 | 21.44 |

low JS and correctly classifying these locations as background, thereby maintaining consistency. This indicates that shared-Gumbel sampling effectively steers the model to select identical tokens in non-edit regions, which can be restored to the source tokens later. (2) Without post-refinement. Removing post-refinement causes ERec to underperform the baseline in background preservation, indicating that reconstruction alone tends to introduce background changes. With post-refinement enabled, ERec can correct such errors and robustly restore background consistency. (3) Without pixel alignment. Disabling pixel alignment degrades background metrics, indicating that our mask-guided alignment can more faithfully preserve textures, consistent with the visual evidence in Figure 3.

**Ablation on post refinement.** We further ablate the mask design within post-refinement, starting from a *base* that uses only shared Gumbel and pixel alignment, and then adding components step by step. (1) JS-only seeds. Using only high-JS locations yields poor performance on both background preservation and edit similarity, indicating that the initial mask is too coarse. (2) + Dilation. Adding 8-connected dilation raises edit similarity to roughly match the baseline, but further harms background preservation, suggesting substantial background leakage in the mask. (3) + Editing-NLL filter. Incorporating an editing-confidence filter (low $NLL^{edit}$) reassigns tokens that the editor is confident to keep (non-edits) back to background, improving background consistency without suppressing true edits. (4) + Reconstruction-NLL safeguard. Finally, adding a reconstruction-deviation safeguard (high $NLL^{rec}$) prevents misclassification when reconstruction drifts, stabilizing the background while preserving edits. Hyperparameter analysis of post-refinement is provided in Appendix D.2.

## 5 CONCLUSION

In this paper, we presented ERec, a finetuning-free, reconstruction-guided approach for background preservation in AR-based image editing. By introducing shared Gumbel sampling across reconstruction and editing pipelines and coupling it with a lightweight post-refinement stage, ERec enforces background consistency while preserving editing fidelity. Extensive experiments demonstrate that ERec achieves robust background preservation without compromising edit quality, establishing a practical and effective framework for AR-based image editing.

## ETHICS STATEMENT

This work focuses on improving background preservation in instruction-based image editing using autoregressive models. Our method, ERec, is finetuning-free and operates only at inference time, without requiring additional data collection or fine-tuning. We restrict our evaluation to publicly available benchmarks (PIE-Bench) and do not introduce new sensitive datasets. While image editing methods may be misused for generating deceptive or harmful content, our contribution is primarily methodological, reducing unintended background alterations rather than enabling novel manipulations. We encourage responsible use of this method within research and creative applications.

## REPRODUCIBILITY STATEMENT

We make every effort to ensure reproducibility. Detailed algorithmic descriptions are provided in Sec. 3, with pseudocode in Algorithms 1–3. Hyperparameters, datasets, and evaluation protocols are documented in Appendix C. Our method introduces no new training and relies on publicly available pretrained backbones (EditAR/LlamaGen). We will release inference code, implementation details, and scripts for evaluation upon publication, enabling others to reproduce our reported results.

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

# APPENDICES

## ACKNOWLEDGMENT OF LLM USAGE

In preparing this manuscript, large language models (LLMs) were employed solely as general-purpose writing aids. Their use was limited to word- and sentence-level polishing, including correcting typos, improving grammar, and refining phrasing. LLMs were not used for research ideation, scientific analysis, generation of results, or interpretation of findings. All scientific concepts, methods, analyses, and conclusions presented in this work are entirely the responsibility of the authors.

## A ALGORITHM

---

**Algorithm 1** ERec Algorithm (Main)

---

**Input:** Source image $\mathbf{X}$; source tokens $\mathbf{c}^{\mathrm{src}} \in \mathcal{V}^N$; prompts $(\boldsymbol{\kappa}^{\mathrm{edit}}, \boldsymbol{\kappa}^{\mathrm{rec}})$; AR decoder $f_\theta$; VQ-VAE decoder Dec; thresholds $(\tau_{\mathrm{JS}}, \tau_{\mathrm{low}}, \tau_{\mathrm{high}})$; seed

**Output:** Edited image $\mathbf{Y}$

1: **for** $i = 1$ to $N$ **do**
   $\boldsymbol{\ell}_i^{\mathrm{rec}} = f_\theta(\mathbf{c}_{<i}^{\mathrm{rec}}; \boldsymbol{\kappa}^{\mathrm{rec}}, \mathbf{c}^{\mathrm{src}}), \quad \boldsymbol{\ell}_i^{\mathrm{edit}} = f_\theta(\mathbf{c}_{<i}^{\mathrm{edit}}; \boldsymbol{\kappa}^{\mathrm{edit}}, \mathbf{c}^{\mathrm{src}})$
2: $\quad (c_i^{\mathrm{rec}}, c_i^{\mathrm{edit}}, \mathbf{p}_i^{\mathrm{rec}}, \mathbf{p}_i^{\mathrm{edit}}) \leftarrow \text{SHAREDGUMBELSTEP}(\boldsymbol{\ell}_i^{\mathrm{rec}}, \boldsymbol{\ell}_i^{\mathrm{edit}}, i, \mathrm{seed})$ ▷ Algorithm 2
3: Reshape $\{\mathbf{p}_i^{\mathrm{rec}}\}, \{\mathbf{p}_i^{\mathrm{edit}}\}$ to per-location $\{\mathbf{p}_{u,v}^{\mathrm{rec}}\}, \{\mathbf{p}_{u,v}^{\mathrm{edit}}\}$ on the $H_\ell \times W_\ell$ grid
4: $\mathbf{Y} \leftarrow \text{POSTREFINEMENT}(\{\mathbf{p}_{u,v}^{\mathrm{rec}}\}, \{\mathbf{p}_{u,v}^{\mathrm{edit}}\}, \mathbf{c}^{\mathrm{src}}, \mathbf{c}^{\mathrm{edit}}, \mathbf{X}, \mathrm{Dec}, \tau_{\mathrm{JS}}, \tau_{\mathrm{low}}, \tau_{\mathrm{high}})$ ▷ Algorithm 3
5: **return** $\mathbf{Y}$

---

---

**Algorithm 2** Reconstruction-guided inference (Sec. 3.2)

---

**Input:** Logits $(\boldsymbol{\ell}_i^{\mathrm{rec}}, \boldsymbol{\ell}_i^{\mathrm{edit}}) \in \mathbb{R}^V$, step $i$, seed

**Output:** Tokens $(c_i^{\mathrm{rec}}, c_i^{\mathrm{edit}})$ and distributions $(\mathbf{p}_i^{\mathrm{rec}}, \mathbf{p}_i^{\mathrm{edit}})$

1: Draw a shared Gumbel vector $\mathbf{g}_i \sim \mathrm{Gumbel}(0,1)^V$ with deterministic key $(i, \mathrm{seed})$
2: $\mathbf{p}_i^{\mathrm{rec}} \leftarrow \mathrm{softmax}(\boldsymbol{\ell}_i^{\mathrm{rec}}); \quad \mathbf{p}_i^{\mathrm{edit}} \leftarrow \mathrm{softmax}(\boldsymbol{\ell}_i^{\mathrm{edit}})$
3: $c_i^{\mathrm{rec}} \leftarrow \arg\max_{k \in \mathcal{V}}\{\ell_{i,k}^{\mathrm{rec}} + g_{i,k}\}; \quad c_i^{\mathrm{edit}} \leftarrow \arg\max_{k \in \mathcal{V}}\{\ell_{i,k}^{\mathrm{edit}} + g_{i,k}\}$
4: **return** $(c_i^{\mathrm{rec}}, c_i^{\mathrm{edit}}, \mathbf{p}_i^{\mathrm{rec}}, \mathbf{p}_i^{\mathrm{edit}})$

---

---

**Algorithm 3** Post Refinement (Sec. 3.3)

---

**Input:** Per-location distributions $\{\mathbf{p}_{u,v}^{\mathrm{rec}}\}, \{\mathbf{p}_{u,v}^{\mathrm{edit}}\}$; source tokens $\mathbf{c}^{\mathrm{src}}$; edited tokens $\mathbf{c}^{\mathrm{edit}}$; source image $\mathbf{X}$; decoder Dec; thresholds $(\tau_{\mathrm{JS}}, \tau_{\mathrm{low}}, \tau_{\mathrm{high}})$

**Output:** Final image $\mathbf{Y}$ and final mask $M$

1: **JS seeding and dilation:** compute $\mathrm{JS}(u,v)$ via Eq. 4.
$$M_{u,v}^{(0)} = \mathbf{1}\left\{ \max_{(a,b) \in \mathcal{N}_8[u,v]} \mathrm{JS}(a,b) \geq \tau_{\mathrm{JS}} \right\}.$$
2: **NLL refinement:** get per-location NLLs via Eq. 6.
$$S_{u,v} = \mathbf{1}\left\{ \mathrm{NLL}_{u,v}^{\mathrm{edit}} \leq \tau_{\mathrm{low}} \text{ or } \mathrm{NLL}_{u,v}^{\mathrm{rec}} \geq \tau_{\mathrm{high}} \right\},$$
$$M_{u,v} = M_{u,v}^{(0)} \wedge (1 - S_{u,v}).$$
3: **Background lock (token space):**
$$c_{u,v}^{\mathrm{edit}} = \begin{cases} c_{u,v}^{\mathrm{src}}, & M_{u,v} = 0, \\ c_{u,v}^{\mathrm{edit}}, & M_{u,v} = 1. \end{cases}$$
4: **Pixel-level residual compositing:**
$$\hat{\mathbf{X}}_{\mathrm{edit}} = \mathrm{Dec}(\mathbf{c}^{\mathrm{edit}}), \qquad \hat{\mathbf{X}}_{\mathrm{src}} = \mathrm{Dec}(\mathbf{c}^{\mathrm{src}}),$$
$$\mathbf{R} = \mathbf{X} - \hat{\mathbf{X}}_{\mathrm{src}}, \qquad M_\uparrow = \mathrm{UpsampleToImage}(M),$$
$$\mathbf{Y} = \hat{\mathbf{X}}_{\mathrm{edit}} + (\mathbf{1} - M_\uparrow) \odot \mathbf{R}.$$
5: **return** $\mathbf{Y}$

---

## B  GUMBEL-MAX IS EXACTLY CATEGORICAL SAMPLING

**Proposition B.1** (Gumbel-max trick (Jang et al., 2016)). *Let $\boldsymbol{\ell} \in \mathbb{R}^V$ denote unnormalized log-probabilities (logits) over a finite set $\mathcal{V} = \{1, \ldots, V\}$, and let $\{g_v\}_{v \in \mathcal{V}}$ be i.i.d. random variables with distribution* $\mathrm{Gumbel}(0, 1)$. *Define*

$$\hat{v} = \arg\max_{v \in \mathcal{V}}\{\ell_v + g_v\}.$$

*Then, for every $j \in \mathcal{V}$,*

$$\mathbb{P}(\hat{v} = j) = \frac{e^{\ell_j}}{\sum_{m \in \mathcal{V}} e^{\ell_m}} = \mathrm{softmax}(\boldsymbol{\ell})_j.$$

*In particular, $\arg\max(\boldsymbol{\ell} + \mathbf{g})$ is exactly a sample from the categorical distribution with probabilities* $\mathrm{softmax}(\boldsymbol{\ell})$.

*Proof.* Let $a_v = e^{\ell_v}$ for each $v \in \mathcal{V}$, and let

$$\zeta = \sum_{m \in \mathcal{V}} a_m$$

denote the normalizing constant. The CDF and PDF of $\mathrm{Gumbel}(0, 1)$ are

$$F(t) = e^{-e^{-t}}, \qquad f(t) = e^{-t} e^{-e^{-t}}.$$

Then

$$\mathbb{P}(\hat{v} = j) = \mathbb{P}\big(\ell_j + g_j \geq \ell_m + g_m, \; \forall m \in \mathcal{V}\big)$$

$$= \int_{-\infty}^{\infty} f(z - \ell_j) \prod_{m \neq j} F(z - \ell_m) \, dz$$

$$= \int e^{-(z - \ell_j)} \exp\Big[-\sum_m e^{-(z - \ell_m)}\Big] dz$$

$$= a_j \int e^{-z} \exp\big[-\zeta e^{-z}\big] dz.$$

With the substitution $u = \zeta e^{-z}$ (so $du = -\zeta e^{-z} dz$ and $e^{-z} dz = -du/\zeta$), the limits change from $u : \infty \to 0$, yielding

$$\mathbb{P}(\hat{v} = j) = \frac{a_j}{\zeta} \int_0^{\infty} e^{-u} \, du = \frac{a_j}{\zeta} = \frac{e^{\ell_j}}{\sum_m e^{\ell_m}}.$$

$\square$

**Corollary B.1** (truncation/top-$k$/top-$p$). *Let $\mathcal{K} \subseteq \mathcal{V}$ be any nonempty candidate subset (e.g., obtained via top-k or nucleus/top-p filtering from $\boldsymbol{\ell}$), and let $\{g_v\}_{v \in \mathcal{V}}$ be i.i.d.* $\mathrm{Gumbel}(0, 1)$. *Define*

$$\hat{v}_{\mathcal{K}} = \arg\max_{v \in \mathcal{K}}\{\ell_v + g_v\}.$$

*Then, for every $j \in \mathcal{K}$,*

$$\mathbb{P}(\hat{v}_{\mathcal{K}} = j) = \frac{e^{\ell_j}}{\sum_{m \in \mathcal{K}} e^{\ell_m}} = \mathrm{softmax}(\boldsymbol{\ell}^{\mathcal{K}})_j.$$

*Proof.* Define masked logits $\tilde{\ell}_v = \ell_v$ for $v \in \mathcal{K}$ and $\tilde{\ell}_v = -\infty$ for $v \notin \mathcal{K}$. Then

$$\arg\max_{v \in \mathcal{K}}\{\ell_v + g_v\} = \arg\max_{v \in \mathcal{V}}\{\tilde{\ell}_v + g_v\}.$$

Applying Proposition B.1 (Gumbel-max trick) to $\tilde{\ell}$ yields

$$\mathbb{P}(\hat{v}_{\mathcal{K}} = j) = \frac{e^{\tilde{\ell}_j}}{\sum_{m \in \mathcal{V}} e^{\tilde{\ell}_m}} = \frac{e^{\ell_j}}{\sum_{m \in \mathcal{K}} e^{\ell_m}},$$

which is the renormalized restriction of $\mathrm{softmax}(\boldsymbol{\ell})$ to $\mathcal{K}$.

$\square$

# C IMPLEMENTATION DETAILS

## C.1 IMPLEMENTATION DETAILS

We use the default configuration of EditAR (Mu et al., 2025). All images are resized to $512 \times 512$ for both training and inference. The VQ-VAE tokenizer has a downsampling ratio of 16, yielding a $32 \times 32$ latent grid ($N = 1024$ tokens). Its codebook size is 16,384 ($16K$) and the codebook embedding dimensionality is 8. The text encoder is Flan-T5-XL (Raffel et al., 2020a; Chung et al., 2024), producing a sequence of 120 embeddings. As the text-to-image autoregressive backbone we use the pre-trained LlamaGen GPT-XL model with 36 layers and model dimension 1280. Classifier-free guidance (Ho & Salimans, 2022) is enabled with scale $\eta = 3$ by default. The reconstruction prompt is fixed as *"reconstruct the image without changes."*

## C.2 EVALUATION AND METRICS

We evaluate on PIE-Bench (Ge et al., 2024), which contains 700 images spanning ten editing types: (0) random editing, (1) change object, (2) add object, (3) delete object, (4) change object content, (5) change object pose, (6) change object color, (7) change object material, (8) change background, and (9) change image style. Each scene includes four categories (animal, human, indoor, outdoor) with balanced sampling. Our method (and all other feed-forward baselines) takes the source image and editing instruction as input and predicts the edited target. Inversion-based approaches use the source image, the source prompt, and the target prompt.

For quantitative comparison, we follow common practice:

- **Structure Distance (DINO-ViT)** (Caron et al., 2021). Cosine distance between DINO-ViT (Caron et al., 2021) global features,

$$\text{StructDist}(X, \hat{X}) = 1 - \frac{\phi(X)^\top \phi(\hat{X})}{\|\phi(X)\|_2 \|\phi(\hat{X})\|_2},$$

  where $\phi(\cdot)$ denotes the DINO-ViT global feature embedding. This metric assesses structural preservation. Lower is better (reported $\times 10^3$).

- **MSE (background)** (Gonzalez & Woods, 2018). Mean squared error averaged over non-edited pixels $\bar{M}$:

$$\text{MSE}_{\text{bg}} = \frac{\sum_{u,v} \bar{M}_{u,v} \|X_{u,v} - \hat{X}_{u,v}\|_2^2}{\sum_{u,v} \bar{M}_{u,v}},$$

  where $\bar{M}$ denotes the complement of the edit mask. Lower is better (reported $\times 10^4$).

- **PSNR (background)** (Gonzalez & Woods, 2018). Peak signal-to-noise ratio on $\bar{M}$, derived from $\text{MSE}_{\text{bg}}$:

$$\text{PSNR}_{\text{bg}} = 10 \log_{10}\big(1/\text{MSE}_{\text{bg}}\big),$$

  with pixel intensities scaled to $[0, 1]$. Higher is better.

- **LPIPS (background)** (Zhang et al., 2018). Perceptual distance from deep features (e.g., SqueezeNet/VGG), averaged over $\bar{M}$. Lower is better (reported $\times 10^3$).

- **SSIM (background)** (Wang et al., 2004). Structural similarity index (luminance/contrast/structure) averaged over $\bar{M}$. Higher is better (reported $\times 10^2$).

- **CLIP score (whole & edit-region)** (Radford et al., 2021). Cosine similarity between image and text embeddings. For the edit-region score, pixels outside $M$ are masked before encoding to emphasize local text alignment. Higher is better (reported $\times 10^2$).

# D MORE EXPERIMENTAL RESULTS

## D.1 TOP-$k$ SAMPLING

Table 5: Comparison under different top-$k$ samplers. Bold marks the better score within each pair. Our proposed ERec integrates well with top-$k$ sampling and consistently improves background preservation over the baseline across different $k$ values.

| TopK | Methods | Structure Distance ↓ | Background Preservation | | | | CLIP Similarity | |
|------|---------|------|------|------|------|------|------|------|
| | | | PSNR ↑ | LPIPS ↓ | MSE ↓ | SSIM ↑ | Whole ↑ | Edited ↑ |
| 1 | Baseline | 28.53 | 23.40 | 81.42 | 162.34 | 78.58 | 23.06 | 20.42 |
| | ERec (Ours) | **13.83** | **35.63** | **34.46** | **46.18** | **99.60** | **23.17** | **20.47** |
| 16 | Baseline | 29.76 | 23.25 | 81.93 | 141.78 | 78.12 | 23.68 | **20.93** |
| | ERec | **13.23** | **34.21** | **41.65** | **48.23** | **98.73** | **23.76** | 20.92 |
| 128 | Baseline | 32.87 | 22.12 | 101.00 | 126.31 | 76.63 | **24.10** | **21.39** |
| | ERec | **18.78** | **31.27** | **54.23** | **51.01** | **96.76** | 24.07 | 21.31 |
| 256 | Baseline | 34.81 | 21.87 | 105.40 | 132.10 | 76.10 | **24.18** | **21.36** |
| | ERec | **23.15** | **29.86** | **64.00** | **62.37** | **95.23** | 24.15 | 21.33 |
| 512 | Baseline | 37.25 | 21.57 | 113.62 | 132.65 | 75.51 | 24.12 | 21.43 |
| | ERec | **23.09** | **30.08** | **62.93** | **55.77** | **95.11** | **24.13** | **21.45** |
| 1K (1024) | Baseline | 37.62 | 21.57 | 115.37 | 141.51 | 75.10 | 24.07 | 21.53 |
| | ERec | **28.12** | **28.87** | **74.97** | **70.93** | **93.45** | **24.10** | **21.54** |
| 2K | Baseline | 39.12 | 21.16 | 126.47 | 148.13 | 74.48 | 24.01 | **21.50** |
| | ERec | **26.21** | **27.99** | **80.19** | **75.11** | **92.56** | **24.09** | 21.48 |
| 4K | Baseline | 40.82 | 21.10 | 127.92 | 149.38 | 74.17 | 23.91 | **21.50** |
| | ERec | **26.40** | **27.88** | **82.13** | **75.05** | **92.15** | **24.05** | 21.46 |
| 8K | Baseline | 42.02 | 20.94 | 129.82 | 156.97 | 73.91 | 24.05 | 21.49 |
| | ERec | **29.36** | **28.44** | **81.17** | **78.18** | **92.55** | **24.20** | **21.58** |
| 16K | Baseline | 41.08 | 21.15 | 127.86 | 138.20 | 73.96 | 24.03 | 21.61 |
| | ERec | **30.80** | **27.96** | **85.86** | **82.54** | **86.60** | **24.24** | **21.63** |

## D.2 HYPER-PARAMETERS ANALYSIS

Table 6: Hyper-parameter analysis on JS threshold in Eq. 5 $\tau_{\text{JS}}$. Setting: Top-$k$=512, $\tau_{\text{low}} = 3.0$ and $\tau_{\text{high}} = 10.0$. Increasing $\tau_{\text{JS}}$ produces a more conservative mask that shrinks the detected edit region, typically lowering CLIP-Edited scores while elevating background-preservation metrics.

| $\tau_{\text{JS}}$ | Structure | Background Preservation | | | | CLIP Similarity | |
|---|---|---|---|---|---|---|---|
| | Distance ↓ | PSNR ↑ | LPIPS ↓ | MSE ↓ | SSIM ↑ | Whole ↑ | Edited ↑ |
| 0.6 | 23.16 | 27.06 | 68.99 | 63.37 | 89.19 | 24.10 | 21.40 |
| 0.7 | 22.32 | 27.46 | 66.11 | 60.33 | 89.74 | 24.10 | 21.39 |
| 0.8 | 20.90 | 28.04 | 61.51 | 55.35 | 90.55 | 24.05 | 21.34 |
| 0.9 | 18.22 | 29.18 | 52.83 | 45.26 | 91.89 | 23.92 | 21.14 |

Table 7: Hyper-parameter analysis on the low threshold $\tau_{\text{low}}$ applied on NLL$^{edit}$ in Eq.7. Setting: Top-$k$=512, $\tau_{\text{JS}} = 0.7$ and $\tau_{\text{high}} = 10.0$. Raising $\tau_{\text{low}}$ makes the classifier more prone to labeling tokens as background, which can erroneously absorb portions of true edit regions into the background mask and thus reduce edit completeness.

| $\tau_{\text{low}}$ | Structure | Background Preservation | | | | CLIP Similarity | |
|---|---|---|---|---|---|---|---|
| | Distance ↓ | PSNR ↑ | LPIPS ↓ | MSE ↓ | SSIM ↑ | Whole ↑ | Edited ↑ |
| 0 | 24.35 | 25.85 | 75.70 | 69.80 | 86.72 | 24.09 | 21.40 |
| 1 | 23.75 | 26.63 | 71.72 | 66.18 | 88.34 | 24.11 | 21.40 |
| 2 | 23.19 | 27.03 | 69.27 | 63.72 | 89.02 | 24.10 | 21.43 |
| 3 | 22.32 | 27.46 | 66.11 | 60.33 | 89.74 | 24.10 | 21.39 |
| 5 | 19.57 | 28.48 | 57.99 | 51.27 | 91.19 | 24.01 | 21.26 |

Table 8: Hyper-parameter analysis on the high threshold $\tau_{\text{high}}$ applied on NLL$^{\text{rec}}$ in Eq.7. Settings: Top-$k$=512, $\tau_{\text{JS}} = 0.7$ and $\tau_{\text{low}} = 3.0$.

| $\tau_{\text{high}}$ | Structure | Background Preservation | | | | CLIP Similarity | |
|---|---|---|---|---|---|---|---|
| | Distance ↓ | PSNR ↑ | LPIPS ↓ | MSE ↓ | SSIM ↑ | Whole ↑ | Edited ↑ |
| 5 | 18.18 | 28.87 | 55.54 | 43.19 | 91.06 | 24.00 | 21.26 |
| 8 | 21.20 | 27.89 | 63.22 | 55.03 | 90.13 | 24.09 | 21.38 |
| 10 | 22.32 | 27.46 | 66.11 | 60.33 | 89.74 | 24.10 | 21.39 |
| 12 | 23.64 | 27.10 | 68.30 | 65.20 | 89.35 | 24.09 | 21.40 |
| 15 | 26.35 | 26.72 | 71.40 | 80.37 | 88.80 | 24.05 | 21.38 |

Table 9: Hyperparameter analysis of the dilation kernel size. Here, kernel size = 3 denotes the default 3×3 8-connected dilation. Larger kernels expand the effective edit region under the same JS threshold $\tau_{\text{JS}}$. Settings: Top-$k$=512, $\tau_{\text{JS}}$=0.7, $\tau_{\text{low}}$=3.0, $\tau_{\text{high}}$=10.0.

| kernel size | Structure | Background Preservation | | | | CLIP Similarity | |
|---|---|---|---|---|---|---|---|
| | Distance ↓ | PSNR ↑ | LPIPS ↓ | MSE ↓ | SSIM ↑ | Whole ↑ | Edited ↑ |
| 3 | 22.32 | 27.46 | 66.11 | 60.33 | 89.74 | 24.10 | 21.39 |
| 5 | 23.38 | 26.96 | 69.65 | 64.23 | 89.05 | 24.10 | 21.40 |

### D.3 MORE QUALITATIVE RESULTS

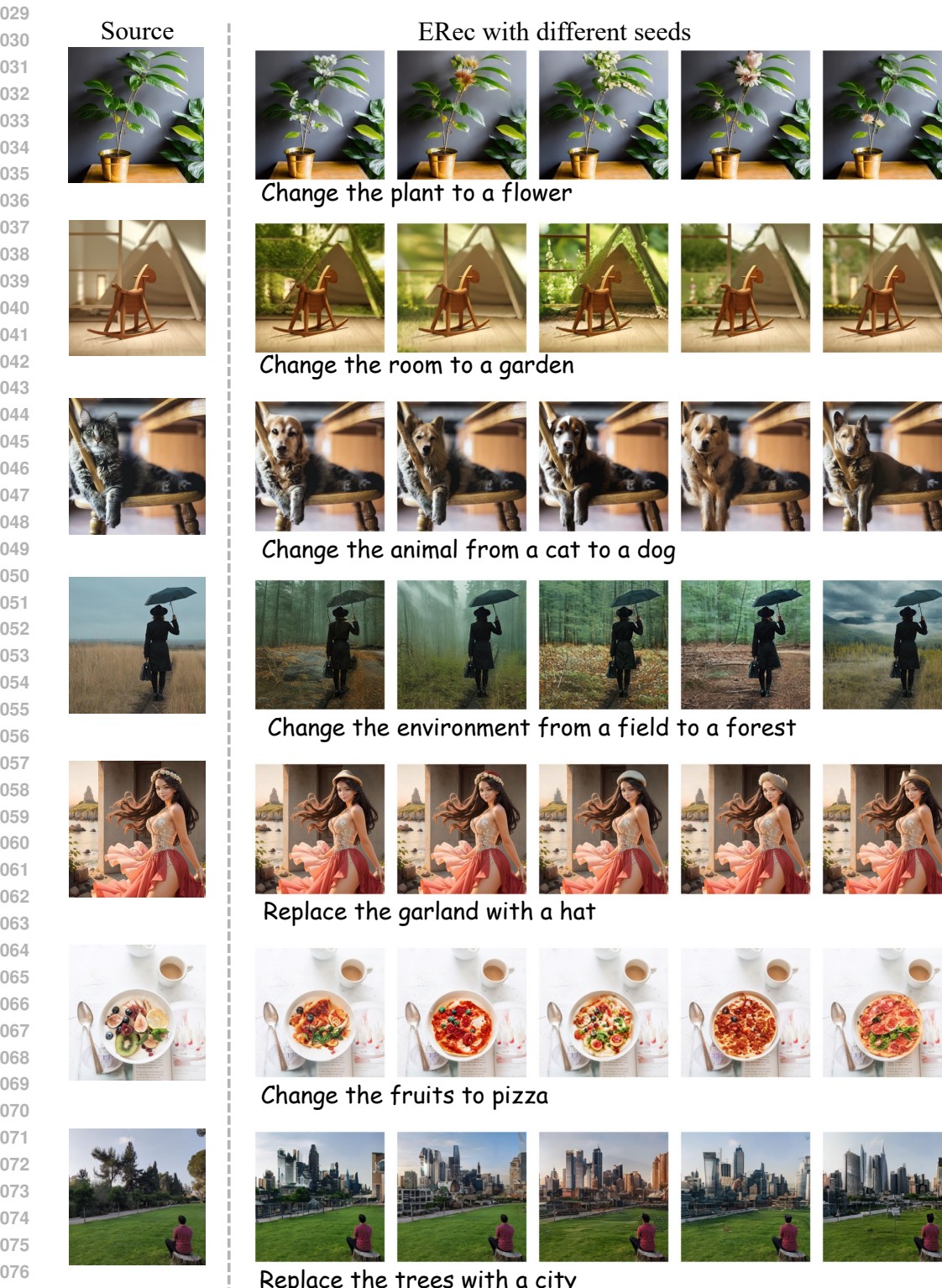

Figure 5: Editing results by ERec with different seeds. Our method does not impair the model's inherent editing ability: diverse styles can still be produced within the edited regions while preserving background consistency.

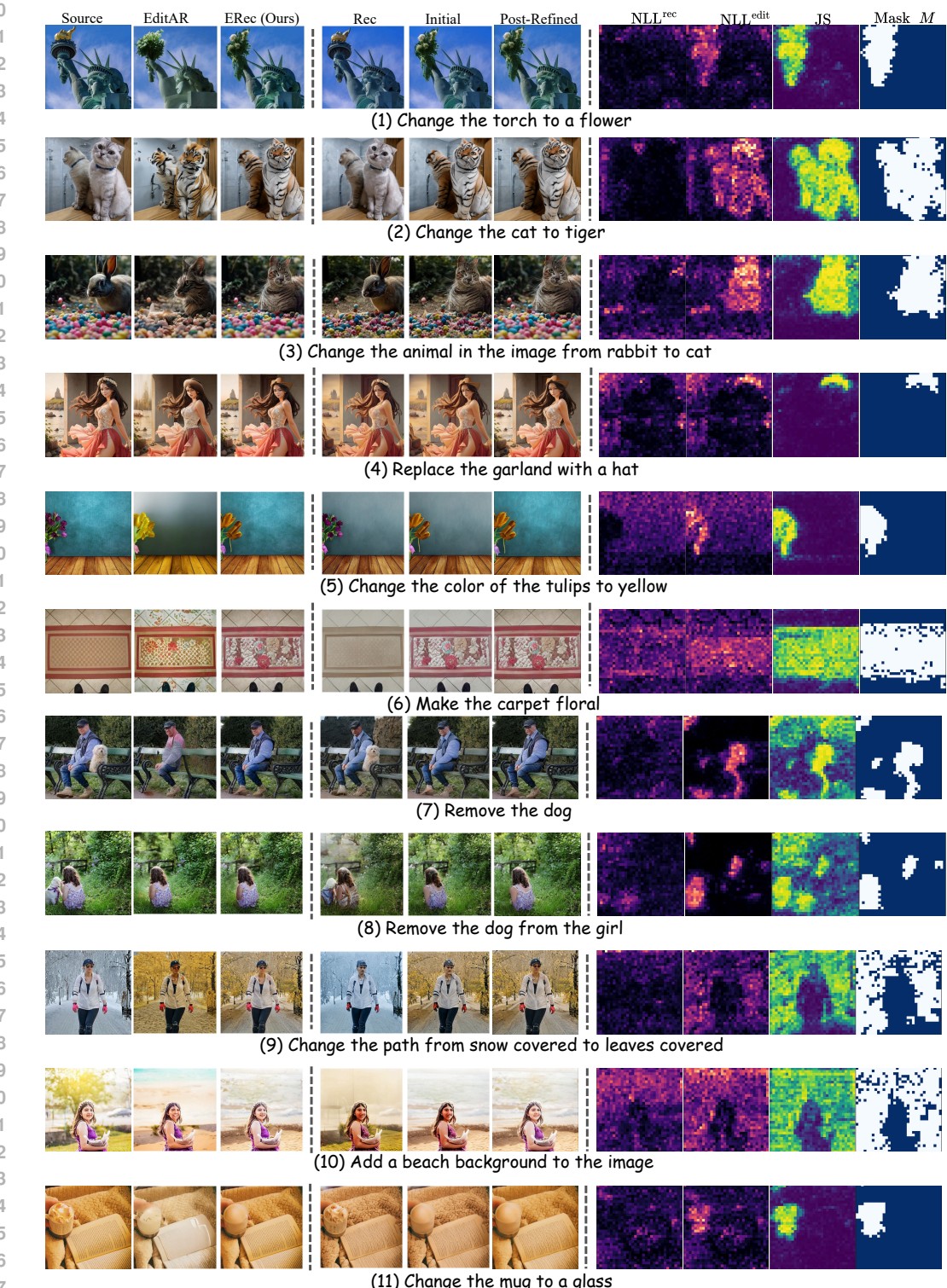

Figure 6: More qualitative results and intermediate process visualization. Correspondence to example cases: (1–4) background preservation; (5) foreground preservation; (6–8) robustness under noticeable reconstruction deviations; (9–10) mask-guided pixel-level restoration of fine details (e.g., facial expressions and text). Higher NLL/JS values indicate potential drift; our post-refinement localizes true edit regions while preserving background consistency.

# E LIMITATIONS

Our method often struggles on the following types of cases: (1) imprecise localization when multiple objects are present or the target object is not salient; (2) edits requiring object or viewpoint transformations; (3) actions that entail large-scale scene changes; and (4) edits that demand external priors or physical reasoning (e.g., Earth illumination). In these scenarios, the output often shows little change or deviates from the instruction. We attribute these failures primarily to the underlying model capacity and the training data distribution. Notably, the baseline EditAR exhibits similar failure modes, typically with worse background preservation.

| Source | EditAR | ERec (Ours) | | Source | EditAR | ERec (Ours) |
|--------|--------|-------------|---|--------|--------|-------------|

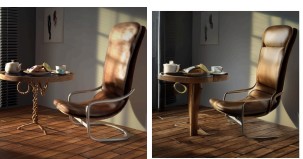 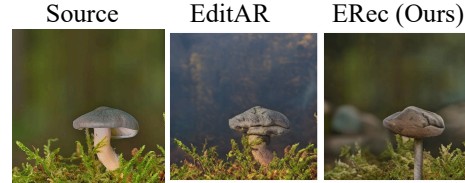

Replace the goat with a horse        change the moss to rocks

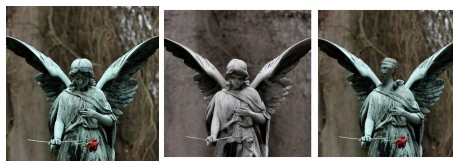

Change the position of the chair to face backwards        Change the perspective of this sculpture from the front to the side

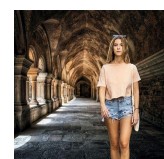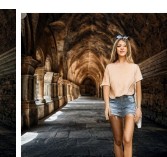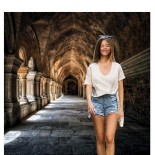 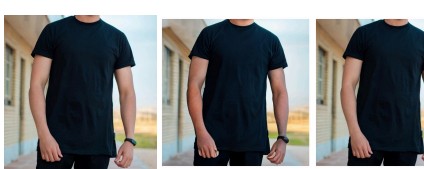

make the woman looking at the right side        make the man's arms cross

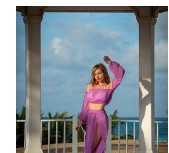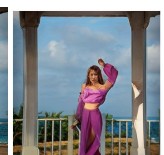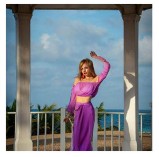 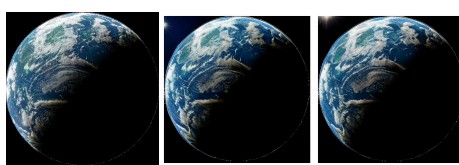

Make the woman's arm being crossed        Change dimly illuminated to sunlit illuminated

Figure 7: Failure cases