# OpenReview forum: "Editing by Reconstruction: Background Preservation for Instruction-based Autoregressive Image Editing"
_ICLR.cc/2026/Conference — ICLR 2026 Conference Withdrawn Submission_

### Official Review · Reviewer_jYtY · 2025-10-20

**Soundness:** 3
**Presentation:** 3
**Contribution:** 2
**Rating:** 4
**Confidence:** 3

**Summary:**

The paper studies instruction-based autoregressive (AR) image editing and targets a persistent issue: unintended changes to non-edited regions due to stochastic token sampling. It proposes ERec (Editing by Reconstruction), a finetuning-free, inference-time framework that runs a reconstruction pass in parallel with the editing pass and injects the same standard-Gumbel noise vector into both logits at every decoding step. This synchronizes sampling so that where editing and reconstruction logits are similar (typically background), token choices align; where they differ (true edits), choices diverge, and editability is preserved. A lightweight post-refinement then localizes edits using Jensen–Shannon (JS) divergence between reconstruction and editing token distributions and per-token negative log-likelihood thresholds, followed by residual pixel compositing to restore background textures. ERec integrates with top-k/top-p sampling without altering multinomial probabilities and requires no model retraining. On PIE-Bench, it yields substantially improved background preservation versus EditAR while keeping CLIP alignment competitive, with modest runtime overhead.

**Strengths:**

* **Originality.** The core idea—record-and-replay sampling via shared Gumbel draws between reconstruction and editing—is simple yet novel for AR editors, directly addressing background drift without architectural changes or training. The multinomial-equivalence argument (and top-k/top-p truncation corollary) is rigorously stated.
* **Quality.** The method is carefully instantiated with explicit objectives and an end-to-end algorithm. Ablations (shared vs independent Gumbel, removal of post-refinement/pixel alignment, “direct replacement” variant) clarify where gains come from. Runtime measurements indicate only a small overhead relative to the baseline.
* **Clarity.** The pipeline is clearly depicted; the post-refinement mask construction (JS seeding, dilation, dual NLL thresholds) is presented step-by-step, with algorithms in the appendix. Figures also visualize diagnostics (JS/NLL maps).

**Weaknesses:**

* **Evaluation scope and external validity.** Results rely primarily on PIE-Bench and automatic metrics; there is no user study, no multi-turn editing, and no evaluation on additional AR backbones beyond EditAR/LlamaGen, limiting generality.
* **Comparative coverage.** Diffusion baselines are included, but AR-specific contemporaries beyond EditAR are not reproduced; some recent training-free AR editing strategies are mentioned in related work but not compared empirically.
* **Trade-offs in semantics.** Gains are strongest on background metrics; CLIP (whole/edited) is largely maintained rather than improved, so the method primarily preserves rather than enhances edit fidelity. A deeper analysis of cases where CLIP degrades would help.
* **Practicality for interactive editing.** Although the overhead is modest, ERec requires a dual-path pass and diagnostics; guidance on early stopping or adaptive step budgets for real-time systems is limited.

**Questions:**

*  **Comparisons within AR.** Related work mentions training-free anchoring and discrete noise inversion; can the authors add these as baselines or discuss why synchronized Gumbel should dominate them theoretically/practically?
*  **User-perceived quality.** Since CLIP may miss human-perceived artifacts, will the camera-ready include a small user study (pairwise preference on background fidelity vs edit fidelity) to support the practical benefit of ERec?

---

### Official Review · Reviewer_qrnx · 2025-10-27

**Soundness:** 3
**Presentation:** 3
**Contribution:** 1
**Rating:** 4
**Confidence:** 5

**Summary:**

The paper improves image editing models on background preservation, i.e., keeping non-editing regions unchanged. To do so, they rely on a pre-trained autoregressive image editing model and tailor a training-free inference algorithm for better background preservation. Experimental results demonstrate the effectiveness of the algorithm.

**Strengths:**

- the paper identifies and mitigates the background preservation issue with image editing models.
- a training-free inference algorithm is proposed for autoregressive image editing models.
- the inference method is tested on popular image editing benchmarks and shows better performance than baselines
- modeling choices of the proposed method are examined through ablation studies.

**Weaknesses:**

- restricted application scope of the proposed method; the algorithm depends on and is tailored for a pre-trained autoregressive image editing model, also implying gains can become marginal as the base model improves.
- though the method improves over the base image editing model, it lags far behind diffusion models like null-text inversion proposed in 2022.
- lacking evaluation on the latest editing benchmarks, such as [EMU-Edit](https://arxiv.org/pdf/2311.10089) and [ImgEdit](https://arxiv.org/pdf/2505.20275).
- overall, the proposed method is interesting and effective, but the contribution is incremental, and the practical use of the method is limited.

**Questions:**

- can you elaborate on $\mathcal{K}_i$ in line 265; is it shared by $\mathbf{c}^{rec}$ and $\mathbf{c}^{edit}$, and where is it derived from?
- any chance you have read [NEP: Autoregressive Image Editing via Next Editing Token Prediction](https://arxiv.org/pdf/2508.06044), which uses the autoregressive paradigm and tackles the same problem, though relying on fine-tuning.

---

### Official Review · Reviewer_UmCf · 2025-10-31

**Soundness:** 2
**Presentation:** 3
**Contribution:** 2
**Rating:** 4
**Confidence:** 4

**Summary:**

ERec is a training-free, dual-path inference add-on to EditAR that runs a reconstruction prompt alongside the editing prompt and injects the same standard Gumbel noise into both paths to synchronize background token choices. It does not alter logits or apply CFG-style reweighting; instead, it derives an editing mask from the distributional gap between the two paths to generate editing pixels without additional supervision (e.g., open-vocabulary segmentation models). In experiments, ERec produces editing performance versus EditAR while preserving background SSIM by 12–13 percentage points at only about a 10% inference overhead.

**Strengths:**

* The proposed ERec runs a reconstruction path and an editing path in parallel with shared Gumbel noise, so non-edited regions pick the same tokens while true edit regions can diverge, and it requires neither retraining nor extra supervision.
* The method automatically produces the editable region with three thresholds: one JS-divergence threshold to seed the region and two per-token NLL thresholds to refine it. This removes the need for external segmentation models and preserves background pixels for the final refinement.
* The approach increases inference from 33.86 s to 37.19 s per image (about 9.8% overhead), while Table 2 shows higher SSIM on background pixels and essentially unchanged CLIP-level edit fidelity.

**Weaknesses:**

**1. Novelty versus Previous Papers for Background Preservation.** The paper does not clearly separate its contribution from diffusion-editing methods [1, 2] that preserve background at the noise or attention level. The authors should clearly show what is new beyond the target for AR baselines (e.g., EditAR).

**2. Limited Evaluation Scope and Practical Relevance of AR Editing.** The evaluation omits strong, widely used editors and recent diffusion-editing models. The authors should compare recent and popular diffusion baselines, such as Qwen-Image [3] and FLUX.1-Kontext [4], UltraEdit [5], In-context Edit [6], and Omniedit [7], and add representative closed-source systems (e.g., GPT-Image-1, Gemini-2.5-Flash) to justify AR’s speed-quality trade-off. Notably, the FLUX.1-Kontext reports H100 inference that is at least three to four times faster than the AR setup described here; if quantitative quality is similar, it is difficult to motivate an AR editing baseline on practical grounds.

**3. Fairness and Transparency of Background Evaluation.** The protocol for measuring background preservation is unclear without ground-truth masks. The authors should define background regions independent of ERec (e.g., external or human-labeled masks), report the accuracy of ERec-induced masks using segmentation metrics such as IoU, and compute background SSIM consistently across all editing methods.

[1] KV-Edit: Training-Free Image Editing for Precise Background Preservation, ICCV 2025.

[2] Early Timestep Zero-Shot Candidate Selection for Instruction-Guided Image Editing, ICCV 2025.

[3] Qwen-Image Technical Report.

[4] FLUX.1 Kontext: Flow Matching for In-Context Image Generation and Editing in Latent Space.

[5] UltraEdit: Instruction-based Fine-Grained Image Editing at Scale, NeurIPS 2024.

[6] In-context edit: Enabling instructional image editing with in-context generation in large scale diffusion transformer.

[7] Omniedit: Building image editing generalist models through specialist supervision, ICLR 2025.

**Questions:**

Q1. Given that FLUX.1-Kontext [4] reports H100 inference at least 3–4× faster, why is AR editing preferable if quantitative edit quality is similar?

Q2. Will you add comparisons against recent diffusion editors such as Qwen-Image [3], FLUX.1-Kontext [4], UltraEdit [5], In-context Edit [6], and Omniedit [7], and include representative closed-source systems to substantiate the speed–quality trade-off?

Q3. How are background regions defined for non-ERec baselines on benchmarks without ground-truth background masks? If an ERec-derived mask is reused across methods, why is this fair, and will you provide external or human-labeled masks to validate background mask accuracy (e.g., IoU)?

Q4. Is there a non-parametric or adaptive alternative to fixed thresholds, for example Otsu-style or percentile rules, or setting \tau in Eq. 5 from a data-dependent statistic such as the mean of the divergence distribution?

Q5. Does the approach generalize to newer AR backbones [8, 9], and can reconstruction or inversion guided sampling, analogous to diffusion editing methods [10, 11], be adapted to text-to-image AR models [9] to enable real-image editing without retraining?

[8] Autoregressive Image Generation without Vector Quantization, NeurIPS 2024.

[9] Infinity∞: Scaling Bitwise AutoRegressive Modeling for High-Resolution Image Synthesis, CVPR 2025.

[10] Null-text Inversion for Editing Real Images using Guided Diffusion Models, CVPR 2023.

[11] Semantic Image Inversion and Editing using Rectified Stochastic Differential Equations, ICLR 2025.

---

### Official Review · Reviewer_oHre · 2025-11-01

**Soundness:** 3
**Presentation:** 3
**Contribution:** 2
**Rating:** 4
**Confidence:** 4

**Summary:**

This paper proposes ERec (Editing by Reconstruction), a finetuning-free method to improve background preservation in instruction-based autoregressive (AR) image editor EditAR. The key idea is to run a reconstruction pass, prompted as “reconstruct the image without changes”. In parallel with the editing pass, and to synchronize their Gumbel-max sampling noise so that background token choices remain consistent. Afterward, a post-refinement stage uses the JS divergence between reconstruction and editing distributions plus NLL-based confidence thresholds to isolate true edited regions, followed by residual pixel compositing to restore background details. Experiments on PIE-Bench show clear improvement in background fidelity.

**Strengths:**

The proposed approach integrates well with EditAR, with acceptable model complexity increase. Post-refinement via JS divergence + NLL thresholds is conceptually simple yet interpretable.

The proposed reconstruction-guided inference is novel and verified as an effective method by experiments.

The paper is well written.

**Weaknesses:**

The proposed ERec model can be viewed as an improvement over EditAR, but it remains tightly constrained within the EditAR framework. Consequently, its technical impact is limited.

The code is not provided for reproducibility checking.

Most experiment comparison is conducted against EditAR. In Table 1 performance comparison is compared with multiple baselines but different T2I models are used.

**Questions:**

My major concern is the limited technical impact of the proposed model, and the baselines are not persuasive enough for performance comparison.

---

### Note · Authors · 2025-11-14

I have read and agree with the venue's withdrawal policy on behalf of myself and my co-authors.